# Rheological and Flow Behaviour of Yolk, Albumen and Liquid Whole Egg from Eggs of Six Different Poultry Species

**DOI:** 10.3390/foods10123130

**Published:** 2021-12-17

**Authors:** Vojtěch Kumbár, Sylvie Ondrušíková, Daniel Trost, Adam Polcar, Šárka Nedomová

**Affiliations:** 1Department of Technology and Automobile Transport (Section Physics), Faculty of AgriSciences, Mendel University in Brno, Zemědělská 1665/1, 613 00 Brno, Czech Republic; xtrost@mendelu.cz (D.T.); adam.polcar@mendelu.cz (A.P.); 2Department of Food Technology, Faculty of AgriSciences, Mendel University in Brno, Zemědělská 1665/1, 613 00 Brno, Czech Republic; sylvie.ondrusikova@mendelu.cz (S.O.); sarka.nedomova@mendelu.cz (Š.N.)

**Keywords:** rheology, flow, velocity, egg, poultry species

## Abstract

Liquid egg products are one of the basic raw materials for the food industry. Knowledge of their rheological and flow behaviour in real technical elements is absolutely necessary for the selection of suitable technological equipment for their processing. In this article, the rheological properties of liquid egg products were determined. Eggs from six different species of poultry are used: domestic hen (*Gallus gallus domesticus*) hybrid *Hisex Brown*; Japanese quail (*Coturnix japonica*); German carrier goose (*Anser anser f. domestica*); domestic ducks (*Anas platyrhynchos f. domestica*); domestic guinea fowl (*Numida meleagris f. domestica*); and domestic turkeys (*Meleagris gallopavo f. domestica*). Liquid egg products showed pseudoplastic behaviour in range of shear strain rates from 0.2 up to 200 s^−1^ and at the temperature of 18 °C. Thus, the flow curves were constructed using the Ostwald-de Waele rheological model, with respect to the pseudoplastic behaviour of liquid egg products. According to the values of the coefficients of determination (*R*^2^), the sum of squared estimate of errors (SSE) and the root mean square error (RMSE), this model was appropriately chosen. Using the consistency coefficient *K*, the flow index *n* and the adjusted equations for the flow rate of technical and biological fluids in standard pipelines, the 3D velocity profiles of liquid egg products were successfully modelled. The values of the Reynolds number of the individual liquid egg products were calculated, and the type of flow was also determined. A turbulent flow has been detected for some liquid egg products.

## 1. Introduction

Eggs have been classified as nature’s original functional food [1]. Recently, there has been a significant increase in efforts to process liquid egg products as a ready-to-eat food product, especially in large quantities [2]. The term processing of liquid egg products is to be understood primarily as their expulsion from shells, filtration, homogenisation, pasteurisation, drying, cooling or freezing. During and after processing, liquid egg products are transported, packaged and stored. It is for these reasons that rheological research on liquid egg products is absolutely justified and necessary.

According to the results of several publications [3,4], the egg albumen and the liquid whole egg are among the non-Newtonian liquids, and the yolk with its rheological behaviour is close to a Newtonian liquid [5,6]. For this reason, the individual description of the flow behaviour of all three liquid egg products (yolk, albumen and liquid whole egg) is important, as in the design of machinery for transport, homogenisation, drying, pasteurisation, etc., no distinction is usually made between the flow properties of the liquid egg products. To describe the rheological behaviour of liquids, flow curves are used, which are often modelled in liquid egg products using power models (Ostwald-de Waele and Herschel-Bulkley model), which contain the consistency coefficient *K*, the flow index *n* and the yield stress *τ*_0_ (Herschel-Bulkley model) [7]. These coefficients also enter into the calculation of external and internal fluid friction, pressure drop, mean and maximum flow velocity, volume and mass flow and the Reynolds number. Furthermore, the type of flow and the shape of the velocity profile can be determined. The determination of viscoelastic properties of egg liquids is also very important for determining their qualities [8]. These are determined using an oscillatory shear test, and the loss modulus and storage modulus are calculated [9].

Almost all of the above works examined only liquid egg products obtained from hen eggs. This work presents a comprehensive overview of the flow behaviour of liquid egg products obtained from hen, quail, goose, duck, guinea fowl and turkey eggs. Goose eggs are very popular in the UK and Asia [10], and duck and guinea fowl eggs are also popular in the Asia region. However, due to the increased health risks associated with their consumption, they are processed into liquid egg products less [11]. In West Africa, guinea fowl is a second source of poultry meat and eggs after hens [12]. From the point of view of food usage of eggs, quails, geese, ducks, guinea fowls and turkeys can be classified as minor species of poultry.

The aim of this work is to determine whether (and what) effect the shear strain rate has on the flow behaviour of liquid egg products (yolk, albumen, liquid whole egg) from eggs of different poultry species, as well as to model flow behaviour of egg liquids using a suitable rheological model. Finally, using the obtained coefficients and appropriate mathematical equations to simulate a non-Newtonian flow in real technical elements, i.e., pipelines. For this research, it was hypothesised that the yolk, albumen and liquid whole egg from the eggs of six species of poultry show different flow behaviours.

## 2. Materials and Methods

### 2.1. Liquid Egg Products

The hen egg samples (40 eggs) came from laying hens of the domestic hen (*Gallus gallus domesticus*) of the *Hisex Brown* hybrid, which were laying at the time of the egg sampling in the 29th week. The laying hens were reared in cage technology and fed a complete feed mixture. The quail egg samples (60 eggs) came from laying hens of the Japanese quail (*Coturnix japonica*), which were laying at the time of egg sampling in the 13th week. The laying hens were reared in cage technology and fed a complete feed mixture. The goose egg samples (30 eggs) came from laying hens of the German carrier goose (*Anser anser f. domestica*), which were laying at the time of egg sampling in the 11th week. The laying hens were kept in the open and fed with a complete feed mixture. The duck egg samples (30 eggs) came from laying hens of the domestic duck (*Anas platyrhynchos f. domestica*), which were laying at the time of egg sampling in the 6th week. The laying hens were kept in the open and fed with a complete feed mixture. The guinea fowl egg samples (30 eggs) came from laying hens of the speckled guinea fowl (*Numida meleagris f. domestica*), which were laying at the time of egg sampling in the 12th week. The laying hens were kept in the open and fed with a complete feed mixture. The turkey egg samples (30 eggs) came from laying hens of the domestic turkey (*Meleagris gallopavo f. domestica*), which were laying in the 3rd week of egg sampling.

In this research, eggs from laying hens of six species of poultry were used-domestic hens, Japanese quail, German laying geese, domestic ducks, spotted guinea fowl and domestic turkeys. An essential part of the experiment was the preparation of the samples themselves. As this is a biological material, it was first necessary to subject all the eggs obtained to a precise inspection, and to only select quality eggs. Eggs that had an atypical shape, a deformed or broken shell or a biological defect were discarded. The individual eggs were beaten by hand. For the preparation of the yolks and albumen, the individual components were carefully separated. The entire egg content was used to prepare the liquid whole egg. The individual liquid egg products were further filtered to remove unwanted components (chalazas, membranes, shell fragments, etc.). These residues need to be removed so that they do not cause problems in the subsequent measurement and do not affect the result, so the operation and filtration is included in this processing. Filtration was performed manually using a conical-shaped colander with a mesh size of 0.7 × 0.7 mm, with constant stirring with a glass rod. By passing through the filter, the egg liquid is not only filtered, but also homogenized. For example, the albumen is formed by four layers, namely the inner dense albumen, the inner thin albumen, the outer dense albumen and the outer thin albumen, the yolk being surrounded by a yolk (vital) membrane [13]. The prepared mixed samples of yolk, albumen and liquid whole egg containing 50 mL were cooled to a temperature of 18 °C, and immediately afterwards subjected to the experiments.

### 2.2. Density Measurement

The density (specific gravity) of the mixed samples of the liquid egg products was measured at the temperature 18 °C using a Densito 30 PX portable digital densitometer (Mettler Toledo, Columbus, OH, USA). The chosen temperature is the legislative upper limit for the storage, transport and sale of eggs. Used densitometer makes it possible to determine the density of the sample in a very short time. The instrument uses the oscillating tube method in combination with an accurate temperature measurement. The device has automatic temperature compensation.

### 2.3. Viscosity and Shear Stress Measurement

A DV–3P rotary viscometer (Anton Paar, Graz, Austria) was used to measure the flow properties of the mixed samples of the liquid egg products, which was equipped with a coaxial cylinder system with a standardised TR8 spindle (according to Anton Paar) and an MX 650 thermostat (AMATEK Brookfield, Middleboro, MA, USA). The used rotary viscometer works on the principle of measuring the moment of force necessary to overcome the resistance of a rotating spindle immersed in the measured material. The rotating spindle is connected via a spring to the motor shaft, which rotates at a defined speed. The angle of rotation of the shaft is measured electronically, and provides accurate information about the position of the shaft, i.e., the spindle. Based on internal calculations, the value of dynamic viscosity or shear stress is directly displayed from the measured values. The shear strain rate was gradually increased from 0.2 to 200 s^−1^. The shear strain rate range was chosen according the literature [4,14]. This range makes it possible to accurately determine and model the flow behaviour and then correctly classify the type (Newtonian, non-Newtonian–with or without yield stress, shear thinning or thickening) of egg liquids. The stable temperature of all liquid egg products in this experiment was also 18 °C.

The egg liquids flow and viscosity curves were modelled using both the Ostwald-de Waele model and the Herschel-Bulkley model. After a thorough analysis, when the yield stress values were 0 Pa (or close) for all samples and in according to publications [15,16,17], the Ostwald-de Waele model was chosen as the most suitable for modelling liquid egg products:(1)$$\tau =K{\dot{\gamma }}^{n}$$

The following applies to the apparent viscosity
(2)$$\eta_{app}=K{\dot{\gamma }}^{n-1},$$
where *K* [Pa·s^n^] is the consistency coefficient and *n* is the dimensionless flow index. These two coefficients are also of considerable physical importance [18]. This is especially true for the flow index *n*, for which the following applies:

0 < *n* < 1 … the fluid behaves non-Newtonian–pseudoplastically (shear thinning),

*n* = 0 … the fluid behaves Newtonian,

*n* > 1 … the fluid behaves non-Newtonian–dilatantly (shear thickening).

### 2.4. Real Technical Elements

To model the flow behaviour of liquid egg products, a real pipeline was selected, which is a typically part of, for example, transport, homogenisation or pasteurisation equipment for the continuous pasteurisation of liquid egg products. In this pipe, there may be changes in pressure, and thus in the flow rate, which may change the type of flow from laminar to turbulent and vice versa. Individual manufacturers use two pipe diameters as a standard-50 mm and 80 mm. The pipe was made of stainless-steel type AISI 304 (1.4301) with a surface roughness up to R_a_ 0.8, which is suitable for the food industry. Stainless steels have a roughness R_a_ of 0.2–0.5 µm after cold rolling to thicknesses up to 4 mm, so they generally do not require polishing unless areas with a roughness higher than R_a_ 0.8 µm are formed during subsequent production operations [19].

Special procedures in Matlab and the equation are used to model the flow velocity of egg liquids in a real technical element (pipeline) [7]
(3)$$v\left(r\right)={\left(\frac{R\Delta p}{2LK}\right)}^{\frac{1}{n}}\left(\frac{nR}{1+n}\right)\left(1-{\left(\frac{r}{R}\right)}^{1+\frac{1}{n}}\right)={\left(\frac{\Delta p}{2LK}\right)}^{\frac{1}{n}}\left(\frac{n}{n+1}\right)\left(R^{\frac{n+1}{n}}-r^{\frac{n+1}{n}}\right),$$
where *v* is the flow velocity at a distance *r* from the longitudinal axis of the pipe and *R* is the inner radius of the pipe having length *L*. Coefficient *K* is the consistency coefficient, *n* is the flow index and Δp is pressure drop.

### 2.5. Statistical Analysis

The measured viscosity and shear stress values depending on the shear strain rate were evaluated and processed using the MATLAB 2018b software (MathWorks, Natick, MA, USA) with the Curve Fitting toolbox. To determine statistical significance between species, an ANOVA test was performed with multiple comparisons using the Tukey’s test in the software Statistica 12 (StatSoft, Tulsa, OK, USA). Statistical testing was performed at a significance level of 95% (*p* < 0.05).

Three statistical indicators were used for the statistical evaluation of the models-the coefficient of determination (*R*^2^), the squared estimate of errors (SSE) and the root mean square error (RMSE). The Fischer-Snedecor test was used to determine the significance of the model, which confirmed that the model was statistically significant (at a significance level of 95%). The statistical significance of the regression parameters of the respective model was verified by means of *t*-tests, which also confirmed the significance of the calculated parameters for the given model (*p* < 0.05).

## 3. Results and Discussion

### 3.1. Density Measurement

Fluid density is an important parameter in modelling fluid flow properties, especially in calculations of the mass flow and energy of a flowing fluid. The density of the liquid egg products was monitored. The individual density values are given in Table 1.

The highest density values were reached by the albumen, followed by liquid whole egg, and the lowest density was reached by the yolk. The density values are given by the composition of the individual egg components [20]. The differences between the density values for the individual poultry species are not statistically significant. Only duck yolk showed about a 2% lower density, which is due to the higher fat content [21].

### 3.2. Viscosity and Shear Stress Measurement and Modelling

To obtain the dependence of the shear stress *τ*, resp. apparent viscosity *η_app_*, on the shear strain rate $\dot{\gamma }$ of the liquid egg products, it is necessary to model (interpolate) the measured data by a suitable function or model, see Equations (1) and (2). Table 2 shows the calculated coefficients of the Ostwald-de Waele model of the liquid egg products from the fresh eggs of six species of poultry at a temperature 18 °C.

Table 2 shows that the rheological Ostwald-de Waele model was very appropriately chosen in accordance with the literature [22,23], as the coefficient of determination *R*^2^ reached values of 0.9165 on average, and the SSE values were 2.6429 on average, which are very good values [24].

For comparison, the flow curves of the fresh yolks are shown in Figure 1, the egg albumen are shown in Figure 2, and the liquid whole egg at 18 °C is shown in Figure 3 for all the monitored poultry species.

The highest values of the shear stress were reached by the duck yolk, goose albumen and turkey liquid whole egg. The world’s most used liquid egg products (hens) were in the lower half of the value of shear stress (130 Pa for yolk, 1.7 Pa for albumen, and 1.8 Pa for LWE at 200 s^−1^), in comparison with the liquid egg products from the eggs of the other five species of poultry (615 Pa for duck yolk, 5.4 Pa for goose albumen and 12 Pa for turkey LWE at 200 s^−1^). The study emphasises the effects of species, which are reflected in changes in rheology. The type of eggs also differs in the representation of individual egg components, when quail and duck eggs have a higher yolk ratio than albumen, compared to hen eggs [7,25]. The proportions of yolk, lipids and water in the eggs of altricial and precocial birds vary considerably. The eggs of altricial birds have the lowest average yolk content (24%), while the eggs of precocial birds have the highest average yolk content (65%). Based on the weight of the egg white, one hen’s egg corresponds to 0.81 duck egg or 5.9 quail eggs, and these weight differences are reflected in the representation of individual egg albumen components, where the types of thin and hard egg white alternate, where each species forms a different representation with partial differences in composition, which also lead to changes in the rheological determination [26]. The nutritional composition, quality and rheology of egg liquids can be influenced also by the composition of the feed mixture [27].

From above, it can be concluded that in the intended processing of the liquid egg products (e.g., goose, duck or turkey) pasteurisation and freezing equipment will have to be dimensioned much more precisely to avoid turbulence and technology failures [28] using modelling of the flow behaviour of the liquid egg products [29,30].

### 3.3. Pipe Flow Velocity Modelling

As a further comparison of the flow behaviour of liquid egg products, it is possible to use three-dimensional modelling of the velocity profiles of the yolks, albumen and liquid whole egg from the eggs of all the monitored poultry species. Figure 4 compares the three-dimensional velocity profiles of the yolks, Figure 5 compares the three-dimensional velocity profiles of the albumen and Figure 6 compares the three-dimensional velocity profiles of the liquid whole egg of all the monitored poultry species using Equation (3) and specific modelling procedures in Matlab. For a comparative example, a pipe with length *L* = 3 m and internal diameter *D* = 80 mm (*R* = *D*/2 = 40 mm) and pressure drop ∆*p* = 300 Pa was used.

As can be seen from Figure 4, Figure 5 and Figure 6, the velocity profiles in the yolks are quite similar–parabolic, suggesting a laminar flow of the liquid egg products [31]. For the albumen and liquid whole egg, flattened shapes appear in velocity profiles, suggesting a turbulent flow [32].

### 3.4. Determination of Flow Type

For an exact decision whether it is a laminar or turbulent flow (or a transition region), it is necessary to calculate the Reynolds number *Re* for the individual cases [7]. Table 3 shows the values of the Reynolds number calculated from Equations (4) and (5) when flowing through the pipe (length *L* = 3 m, pressure drop Δ*p* = 300 Pa, inner diameter *D* = 50 mm and 80 mm) for the raw liquid egg products from eggs of all the monitored poultry species at a temperature of 18 °C.
(4)$$Re=\frac{D^{n}v_{s}{}^{\left(2-n\right)}\rho }{8^{n-1}K}{\left(\frac{4n}{1+3n}\right)}^{n},$$
where $v_{s}$ is the mean flow velocity calculated by the relationship
(5)$$v_{s}=\frac{Q_{V}}{S}=\frac{\frac{\pi nR^{3}}{3n+1}{\left(\frac{R\Delta p}{2LK}\right)}^{\frac{1}{n}}}{\pi R^{2}}=\frac{nR}{3n+1}{\left(\frac{R\Delta p}{2LK}\right)}^{\frac{1}{n}},$$
where $Q_{V}$ is the volume flow and *S* is the cross-sectional area of the pipe.

The numerical limit of the laminar flow of liquid egg products is based on Equation (6) and the numerical limit of the turbulent flow of liquid egg products is based on Equation (7). The transition area between laminar and turbulent flow of non-Newtonian power fluids must be determined for each fluid separately [33]. For example, if a power-law fluid has a flow index of *n* = 0.5, the transition region of the flow falls within the range of values of the Reynolds number *Re* from 2675 to 3575 [34]:(6)Re ≤3250−1150n,
(7)Re ≥4150−1150n.

Table 4 lists the flow types, which are determined according to the intervals obtained from Equations (6) and (7).

It can now be stated that turbulent flow was achieved under the given conditions (*L* = 3 m, pressure drop Δ*p* = 300 Pa, inner diameter *D* = 50 mm and 80 mm, *T* = 18 °C) for the fresh hen albumen and liquid whole egg only, the fresh duck albumen and the fresh guinea fowl albumen. In the other cases, the fresh liquid egg products had a laminar flow. Turbulent flow in real pipes is problematic, and can lead to wear [35], vibrations [36], and gradually to the failure of the technology or system. Turbulence must be prevented by choosing a suitable pipe geometry, by changing the temperature (it affects viscosity [37]) or by regulating the pressure drop (it affects volume flow [24]). Therefore, it is very appropriate to use process modelling and simulation in food technologies to prevent production and energy losses [38].

## 4. Conclusions

The highest differences in density of egg liquids were measured in the yolks. In particular, the duck yolk showed the lowest density value. The other liquid egg product values were almost similar to the density of the liquid egg products of hens and quails, although statistically significant differences were found (*p* < 0.05). The highest values of the shear stress of individual groups of egg liquids were reached by the duck yolk, goose albumen and turkey liquid whole egg.

The velocity profiles in the yolks are quite similar–parabolic (laminar flow). For the albumen and liquid whole egg, flattened shapes appear in the velocity profiles (indicate turbulent flow). According to the Reynolds number, turbulent flow was achieved for the hen albumen and liquid whole egg, the duck albumen and the fresh guinea fowl albumen. In the other cases, the fresh liquid egg products had a laminar flow.

From the above results, it can be stated that the tested hypothesis was confirmed, as differences were found in the rheology and flow behaviour of the individual liquid egg products from the six different species of poultry. The obtained coefficient of rheological models can be applied in conventional technical practice in the design of food technological equipment and in the current trends in the food industry–processing eggs of minor species of poultry in a place far from their laying (reduction of carbon footprint).

## Figures and Tables

**Figure 1 foods-10-03130-f001:**
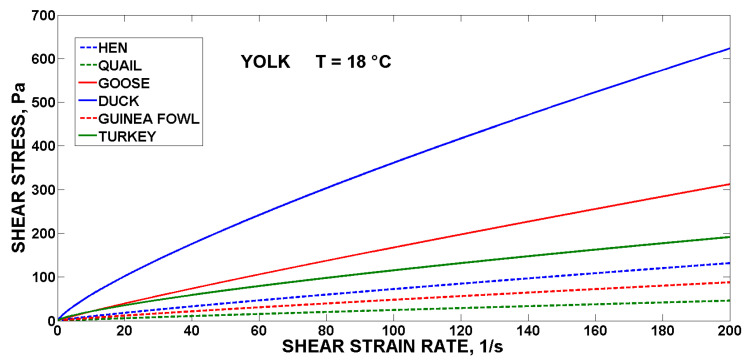
Flow curves of the yolks of the six species of poultry.

**Figure 2 foods-10-03130-f002:**
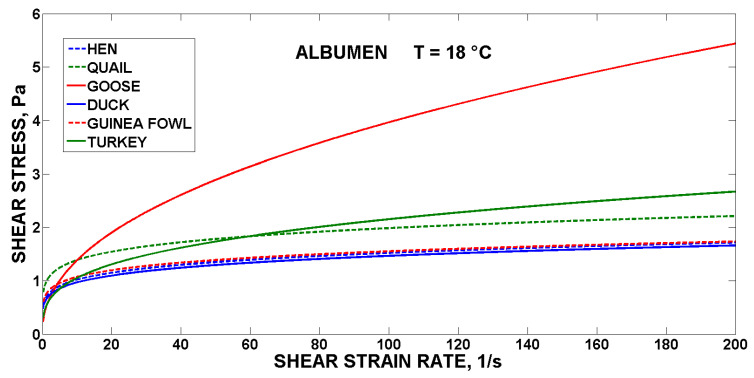
Flow curves of the albumen of the six species of poultry.

**Figure 3 foods-10-03130-f003:**
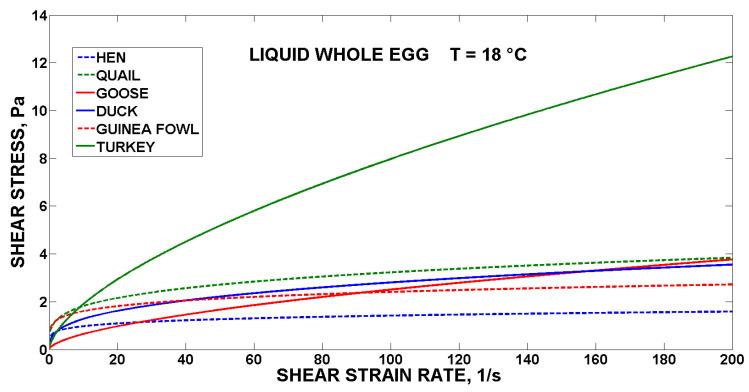
Flow curves of the liquid whole egg of the six species of poultry.

**Figure 4 foods-10-03130-f004:**
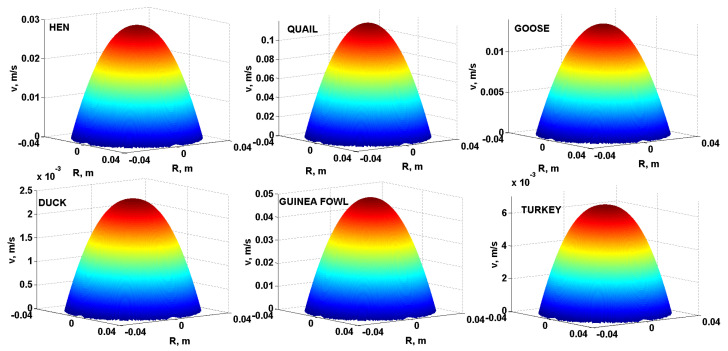
Three-dimensional velocity profiles of the yolks of all six species of poultry.

**Figure 5 foods-10-03130-f005:**
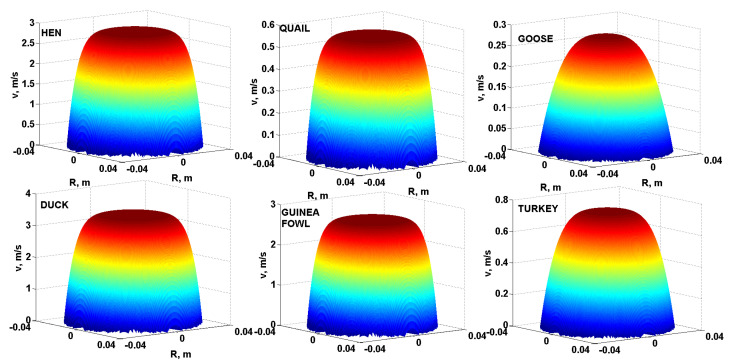
Three-dimensional velocity profiles of the albumen of all six species of poultry.

**Figure 6 foods-10-03130-f006:**
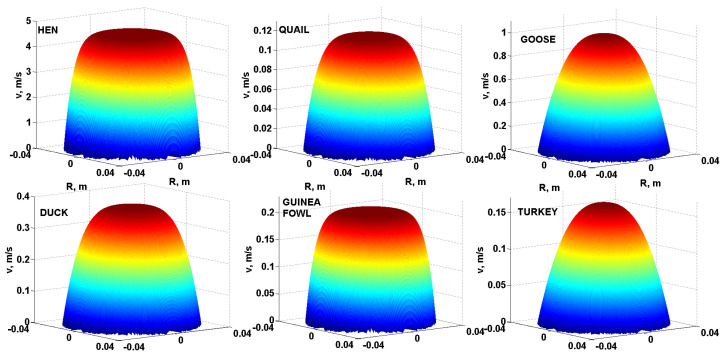
Three-dimensional velocity profiles of the liquid whole egg of all six species of poultry.

**Table 1 foods-10-03130-t001:** Values of the density of the liquid egg products of the individual species of poultry (the number of repetitions *N* = 5, the results are given in the form mean ± standard deviation).

Poultry Species	Density *ρ*, kg·m^−3^		
	Yolk	Albumen	LWE
Hen	1027.7 ^c,d^ ± 0.70	1038.1 ^j^ ± 0.96	1033.2 ^f,g^ ± 0.81
Quail	1021.0 ^b^ ± 0.57	1038.4 ^j^ ± 0.75	1031.7 ^e,f^ ± 0.90
Goose	1031.9 ^e,f^ ± 0.42	1035.5 ^h,i^ ± 0.62	1033.8 ^g,h^ ± 0.60
Duck	1009.3 ^a^ ± 0.54	1038.2 ^j^ ± 0.55	1029.1 ^d^ ± 0.73
Guinea fowl	1027.4 ^c,d^ ± 0.71	1042.6 ^k^ ± 0.67	1035.7 ^i^ ± 0.63
Turkey	1026.4 ^c^ ± 0.67	1037.7 ^j^ ± 0.50	1031.4 ^e^ ± 0.79

The same letters in the compared groups indicate that no statistically significant difference was found between these groups. Abbreviation: LWE—Liquid Whole Egg.

**Table 2 foods-10-03130-t002:** Ostwald-de Waele model coefficients.

Poultry Species	Egg Liquid	Ostwald-de Waele Model				
		*K*, Pa·s ^n^	*n*, -	*R* ^2^	SSE	RMSE
Hen	Yolk	1.3594	0.8632	0.9996	1.1240	0.2737
	Albumen	0.6824	0.1740	0.8311	0.3516	0.1438
	LWE	0.6696	0.1631	0.8789	0.3932	0.1521
Quail	Yolk	0.3769	0.9065	0.9965	7.0460	0.6438
	Albumen	0.9670	0.1561	0.8124	1.2170	0.2675
	LWE	1.0201	0.2501	0.7583	3.5930	0.4597
Goose	Yolk	2.6780	0.8982	0.9990	3.2030	0.4964
	Albumen	0.4831	0.4570	0.9415	1.7940	0.3249
	LWE	0.1674	0.5877	0.9381	0.8657	0.2257
Duck	Yolk	9.6730	0.7861	0.9960	10.550	1.0270
	Albumen	0.6389	0.1802	0.8801	0.7986	0.2167
	LWE	0.5832	0.3410	0.8647	1.7160	0.3177
Guinea fowl	Yolk	0.8542	0.8745	0.9976	6.0971	0.6173
	Albumen	0.7387	0.1610	0.8930	0.4580	0.1641
	LWE	1.0720	0.1758	0.8526	1.4450	0.2915
Turkey	Yolk	3.9770	0.7310	0.9985	3.3050	0.5042
	Albumen	0.5139	0.3109	0.8712	1.7240	0.3184
	LWE	0.4557	0.6214	0.9880	1.8801	0.3325

Coefficient is statistically significant for confidence level *p* < 0.05. Abbreviations: LWE—Liquid Whole Egg; *R*^2^—coefficient of determination; SSE—sum of squared estimate of errors; RMSE—root mean square error.

**Table 3 foods-10-03130-t003:** Values of the Reynolds number of the flowing liquid egg products.

Poultry Species	Pipe Diameter	Reynolds Number *Re*		
		Yolk	Albumen	LWE
Hen	80 mm	0.93	2.03 × 10^4^	5.36 × 10^4^
	50 mm	0.20	57.12	104.80
Quail	80 mm	15.42	830.48	29.39
	50 mm	3.42	1.26	0.43
Goose	80 mm	0.20	123.45	1387.41
	50 mm	0.04	9.86	175.16
Duck	80 mm	0.01	2.88 × 10^4^	257.77
	50 mm	0.001	97.61	10.23
Guinea fowl	80 mm	2.68	1.86 × 10^4^	105.86
	50 mm	0.57	33.83	0.32
Turkey	80 mm	0.05	1075.40	36.29
	50 mm	0.01	32.69	5.00

Abbreviations: LWE—Liquid Whole Egg.

**Table 4 foods-10-03130-t004:** Types of flow of the liquid egg products of all the poultry species.

Poultry Species	Pipe Diameter	Yolk	Albumen	LWE
Hen	80 mm	L	T	T
	50 mm	L	L	L
Quail	80 mm	L	L	L
	50 mm	L	L	L
Goose	80 mm	L	L	L
	50 mm	L	L	L
Duck	80 mm	L	T	L
	50 mm	L	L	L
Guinea fowl	80 mm	L	T	L
	50 mm	L	L	L
Turkey	80 mm	L	L	L
	50 mm	L	L	L

Abbreviations: LWE—Liquid Whole Egg; L—Laminar flow; T—Turbulent flow.

## Data Availability

The data presented in this study are available on request from the corresponding author.

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
