# Peer review of "Rheological and Flow Behaviour of Yolk, Albumen and Liquid Whole Egg from Eggs of Six Different Poultry Species"

_foods, 2021, doi:10.3390/foods10123130_

Round 1
Reviewer 1 Report
In this article the rheological properties (flow curves and type of flow) of liquid egg products (yolk, albumen, and liquid whole egg) were measured. Six different poultry species were chosen as a source of the tested materials. The concept of the research is interesting, however, the very laconic way of developing the results and practically no results’ discussion means that the article does not represent a high scientific value. More specific comments are given below.
Section 2.1. Please start this section with paragraph provided in lines 89-105.
Line 76: Replace “is the preparation” with “was the preparation”.
Line 82: How were the egg products filtrated? Please specify the way of filtrating.
Section 2.2. I find the information in lines 111-116 as redundant.
Section 2.3: Please justify the choice of such a shear strain rate range.
Section 3: Remove lines 150-152.
Section 3.1: Please provide the average values in Table 1 with an upper index denoting statistically significantly different values and provide appropriate explanation note under Table 1.
Section 3.2: Information in lines 168-179 should be given in section 2.3. The results are not discussed. For example, please provide discussion on the effect of the particular component of the egg products on the rheological behaviour of these materials. Line 205: What do authors mean by “much more carefully” – provide any examples for the statement in question. Lines 201-203: No numerical data are provided (also as for the references). What is a conclusion? Line 188: Reference 22 is not appropriate here (inappropriate self-citation by authors).
Section 3.3: Information in lines 213-219 should be given in section 2.4. Complete the explanation of the symbols (K, n, Dp).
Section 3.4: What is a difference between the statements given in lines 231 and 243? Reconsider the calculations (values) in line 249. Lines 256-260: This is a repetition of the data presented in Table 4. There is lack of discussion of the results.
Conclusions are not conclusions indeed. In this section the results and finding are repeated. For example, the text in lines 267-271 is almost identical with this given in lines 199-203. The following sentence (lines 271-274) is identical to that given in lines 203-206. The texts in lines 274-277 and 277-281 are repetitions of these provided in lines 226-229 and 256-261, respectively. Lines 269-271: What about it?
Reference 3: There is typing error in the first author’s name (should be Juszczak).
Please review the manuscript in respect of English correctness (i.e. lines 61-62; 65-69; 198-202).
Author Response
Dear reviewer,
Thank you for your useful comments, which have been incorporated into the text. This certainly raised his scientific level. The response to comments is attached in a separate document and the manuscript is modified for clarity using change tracking mode.
Best regards,
Vojtech Kumbar

Reviewer 2 Report
Manuscript Number: foods-1478411
Title: Rheological and Flow Behaviour of Yolk, Albumen and Liquid Whole Egg from Eggs of Six Different Poultry Species
Comments:
Having thoroughly reviewed the above manuscript, the reviewer suggest that this manuscript should be returned with a major revision. This manuscript investigated the flow behaviors of liquid egg products from laying hen eggs of six different species of poultry. The obtained results show some findings about the effect of poultry species on the flow properties of liquid egg products. However, the present experimental design seems to be relatively simple, especially the measurements of the rheological behaviors of LWE. The authors only determined the flow curves of liquid egg products using a rotary viscometer and then analyzed the corresponding flow index. The rheological viscoelastic properties of liquid egg products are also very important factors for determining their qualities, which are not carried out and even not mentioned in the present manuscript. The language of this manuscript needs to be improved since there are some grammar errors and spelling mistakes.
M&M section, why the temperature of 18 °C was selected for density and flow measurements?
For all Tables, a detailed variance analysis is required for these data.
For the flow modelling liquid egg products, what is the difference between the Ostwald–de Waele and Herschel–Bulkley model? The latter appears to be more commonly used for modelling of flow curves.
Authors are also suggested to give a discussion about the underlying reason for the effect of poultry species and egg fractions on their flow behaviors.
The abstract and conclusion sections are required to be improved and reorganized to clearly show the main findings of this manuscript to readers. The content shown in the present conclusion is overstated to some extent.
Author Response

(The authors gave the same response as above.)

Round 2
Reviewer 1 Report
The authors addressed the comments of the reviewer and improved the manuscript, increasing its quality.
Reviewer 2 Report
The revised manuscript has fully addressed my comments.